# *Staphylococcus aureus* Keratitis: Incidence, Pathophysiology, Risk Factors and Novel Strategies for Treatment

**DOI:** 10.3390/jcm10040758

**Published:** 2021-02-13

**Authors:** Jason W. Lee, Tobi Somerville, Stephen B. Kaye, Vito Romano

**Affiliations:** 1School of Medicine, University of Liverpool, Liverpool L69 3GE, UK; J.W.Lee@liverpool.ac.uk; 2Department of Eye and Vision Science, University of Liverpool, Liverpool L7 8TX, UK; Tobi.Somerville@liverpool.ac.uk (T.S.); S.B.Kaye@liverpool.ac.uk (S.B.K.); 3St Paul’s Eye Unit, Royal Liverpool University Hospital, Liverpool L7 8XP, UK

**Keywords:** keratitis, microbial keratitis, *Staphylococcus aureus*, pathophysiology, novel therapy, therapeutics, treatment

## Abstract

Bacterial keratitis is a devastating condition that can rapidly progress to serious complications if not treated promptly. Certain causative microorganisms such as *Staphylococcus aureus* and *Pseudomonas aeruginosa* are notorious for their resistance to antibiotics. Resistant bacterial keratitis results in poorer outcomes such as scarring and the need for surgical intervention. Thorough understanding of the causative pathogen and its virulence factors is vital for the discovery of novel treatments to avoid further antibiotic resistance. While much has been previously reported on *P. aeruginosa*, *S. aureus* has been less extensively studied. This review aims to give a brief overview of *S. aureus* epidemiology, pathophysiology and clinical characteristics as well as summarise the current evidence for potential novel therapies.

## 1. Introduction

Keratitis is a sight-threatening disease [1], with the majority of cases attributed to infective causes [2] such as bacteria, viruses, fungi and protozoa [3]. Within infectious keratitis, bacteria are the most common causative pathogens [3]; among these, *Staphylococcus aureus* (*S. aureus*), *Pseudomonas aeruginosa* (*P. aeruginosa*), and coagulase-negative staphylococci (CNS) are the most frequently identified causative organisms [4]. Resistant bacterial keratitis can severely impact quality of life due to the requirement for prolonged and more intensive treatment. It is also associated with poorer outcomes such as corneal scarring [5], which leads to reduced visual acuity and greater likelihood of requiring surgical intervention [6]. Considering the importance of bacterial keratitis as a worldwide cause of vision loss, there is an increasing demand for more effective antibiotic regimens or alternative treatments [7]. Thorough understanding of the causative pathogen and its virulence factors is vital for the discovery of novel treatments to avoid further antibiotic resistance. While much research related to *P. aeruginosa* has been conducted, *S. aureus* has been less extensively studied. This review will provide a brief overview of *S. aureus* epidemiology, pathophysiology and clinical characteristics as well as summarise the current evidence for potential novel therapies. 

## 2. *Staphylococcus aureus* Keratitis—Incidence

Studies in England have demonstrated variation in the trends of bacterial keratitis. Ting et al. demonstrated an overall increase in the incidence of Gram-positive keratitis in the North East of England alongside a decrease in the incidence of Gram-negatives such as *P. aeruginosa* over 10 years [8]. On the other hand, another study conducted at Manchester Royal Eye Hospital found a decreasing trend of Gram-positives but a statistically significant increase in *S. aureus* cases over 12 years [9]. Tavassoli et al. did not find a statistically significant trend in any of the microorganisms cultured over a period of 11 years in the South West of England [10]. All three of these studies found that *S. aureus* was not the most common cultured Gram-positive; this was instead a CNS species such as *S. epidermidis*. It is important to note that the pathogenic role of CNS species in immunocompetent individuals has been debated for a long time, and it still remains a challenge to distinguish between clinically significant pathogenic isolates and commensals or contaminants [11].

The reporting of MRSA (methicillin-resistant *Staphylococcus aureus*) was not always included, and trends in incidence can vary [12,13,14]. A San Francisco (CA) study reported *S. aureus* bacterial keratitis as being the most common bacteria amongst Gram-positive cultures [15]. This study also found an increasing trend in MRSA incidence over 19 years, with MRSA contributing to 25% of *S. aureus* cultures. A four-year Pennsylvania study showed similar results, but demonstrated a higher proportion of MRSA cases within *S. aureus* keratitis (34.4%) [16]. Only one of the UK studies mentioned above reported MRSA incidence; two cases were found over 10 years [8]. Two studies from Sydney, Australia, report similar findings to the UK studies, with coagulase-negative staphylococci being the most commonly discovered bacteria and low rates of MRSA within *S. aureus* isolates [17,18]. Meanwhile, studies from Mexico, Taiwan and South India confirm a statistically significant increase in MRSA incidence over time [19,20,21].

## 3. Staphylococcus Aureus Keratitis—Pathophysiology

*S. aureus* is a Gram-positive, coagulase-positive cocci [22]. It exists as a commensal microorganism found throughout the skin and mucosa of the human body [23]. It is implicated in a wide range of different systemic diseases such as soft tissue infections and sepsis [23]; it is also a common pathogen in ocular infectious diseases. On the ocular surface, *S. aureus* has a range of different virulence factors which enable it to resist the host immune defences and cause disease.

### 3.1. Overcoming the Tear Film and Adhesion to the Cornea

The tear film is the main defence system within the eye and contains a collection of antibacterial proteins [24], including surfactant protein D (SP-D). SP-D is able to bind to surface lipids and cause the aggregation of bacteria, providing an effective method of clearance [25]. Zhang et al. have demonstrated that mice who are deficient in SP-D are more vulnerable to *S. aureus* infection, and that cysteine proteases secreted by *S. aureus* can interfere with the action of SP-D and help it to evade this aspect of host defence [25]. A cysteine protease inhibitor was demonstrated to improve the antimicrobial activity of SP-D and may therefore be a possible therapeutic target [25]. One of the cysteine proteases, staphopain A, has also been demonstrated to have a role in facilitating the adhesion and invasion of *S. aureus* into the corneal epithelial cell barrier via increased fibronectin binding [26]. 

### 3.2. Alpha-Haemolysin Toxin

*S. aureus* secretes alpha-haemolysin toxin, which has shown to interfere with the corneal epithelial wound healing process, as well as, promote pathogen invasiveness within the inner layers of the cornea [27]. Antibiotics typically eradicate bacteria but not their toxins [28]. Therefore, targeting toxins like alpha-haemolysin may prove to be an effective therapeutic approach that can be used as an adjunct to antibiotics. Targeting virulence factors rather than the microorganisms themselves is also less likely to lead to resistance since the act of eradicating susceptible bacteria and allowing resistant bacteria to thrive is a crucial aspect of how resistance to antibiotics develops [29]. 

### 3.3. Biofilm Formation and Contact Lenses

*S. aureus* is also capable of forming a biofilm on the cornea (Figure 1) and on contact lenses [30,31]. A biofilm consists of a secreted extracellular polymeric substance which protects bacterial populations from host defences. It also contributes to antibiotic resistance: the bacteria are exposed to suboptimal concentrations of any drugs [32], resulting in a response of biofilm thickening by the surviving population. The main gene contributing to biofilm formation in *S. aureus* is the ica gene [30], which is responsible for the secretion of extracellular polymeric substance. Studies have shown that staphylococci isolates can produce biofilms more easily in vitro compared to *P. aeruginosa* due to faster adherence to the surface [33]. The development of novel therapies must consider addressing the biofilm as a possible target, or as a factor determining whether a drug can penetrate through to the underlying bacteria. 

### 3.4. MRSA and Panton–Valentine Leukocidin

In addition to biofilms conferring general antibiotic resistance, *S. aureus* has a notorious reputation for its inherent resistance to beta-lactam antibiotics. These infections, known as MRSA, originated within hospitals after the introduction of penicillin in 1942, but later transitioned into being prevalent within the community after a third wave of antibiotic resistance [22]. There is an overall rise in the number of community- and hospital-acquired MRSA infections [34], which makes it an increasing concern as a cause of bacterial keratitis. The majority of community-acquired MRSA strains secrete the toxin Panton–Valentine leukocidin (PVL), which is otherwise quite rare in methicillin-susceptible *S. aureus* (MSSA) [22]. PVL has cytotoxic activity against many different types of immune cells, but murine model studies have shown that its pathogenic role may be dependent on the specific strain of *S. aureus* [34]. While its virulence has not been as extensively studied as the toxin alpha-haemolysin, one study concluded that PVL-positive *S. aureus* was associated with worse clinical outcomes and more surgical interventions [35].

## 4. Risk Factors, Clinical Presentation and Diagnosis

Microbial keratitis has numerous risk factors, with the most significant being improper contact lens hygiene. Other common risk factors include ocular trauma, ocular surgery and ocular surface disease [36,37]. Ocular surface disease includes meibomian gland dysfunction, blepharitis, atopy, dry eye syndrome, dacryocystitis and ectropion [36,37,38]. The risk factors of *S. aureus* keratitis appear to be similar to those caused by other bacteria, but evidence is limited for studies focusing on *S. aureus*, and further work should be carried out to determine whether any risk factors for microbial keratitis are particularly important in the development of *S. aureus* infection. Ong et al. found that the prevalence of ocular surface disease was an especially important risk factor for MRSA keratitis compared to MSSA [39]. 

Microbial keratitis is also a recognised severe complication of the corneal cross-linking procedure (CXL). Particularly in this context, there is evidence that the use of a bandage contact lens and topical corticosteroids before the epithelium has healed post-procedure can increase the risk of microbial keratitis significantly [40]. Furthermore, if the patient undergoing CXL has other eye conditions that compromise recovery of the epithelium, such as atopic conjunctivitis or diabetes mellitus, this can also contribute to risk of post-operative microbial keratitis following CXL [40].

The main symptoms that present during bacterial keratitis are non-specific. These include a red, painful eye with excessive tearing and, occasionally, vision loss [29]. Rarer symptoms include photophobia, lid and conjunctival oedema [41]. Bacterial keratitis occurs because of a breach in the corneal epithelium and therefore often takes the form of a corneal ulcer [42]. Corneal ulcers can range from superficial erosions, which only involve loss of the corneal epithelium, to a deeper corneal ulcer that invades the stroma and, if left untreated, can lead to corneal perforation [42]. Figure 2 illustrates the layers of the cornea that can be affected by corneal ulcers. 

Although not always reliable, the clinical presentation may provide a clue as to the causative pathogen. The hallmark presentation of staphylococcal infection is not as clear as other causative bacteria such as *Pseudomonas* spp. (ring abscess) [32], but *S. aureus* infection is thought to present with a greater degree of stromal infiltration than *S. epidermidis* [41]. Figure 3 illustrates the clinical appearance of keratitis caused by *S. aureus* after a full-thickness corneal graft. *S. aureus* has been associated with recurrent bacterial keratitis that causes scarring following each episode [43]. Although scarring is an undesirable complication of keratitis that can lead to vision loss, studies have demonstrated the beneficial role of the scarring process: the blockade of immune-mediated pathways responsible for scarring, such as CXCR2, result in overwhelming bacterial infection [44]. This suggests, therefore, that finding the right balance in an effective therapeutic strategy remains a challenge, especially when immunomodulating adjuncts are used.

Studies have demonstrated that the bacteria-host interaction is complex. Evidence suggests that patients with recurrent keratitis have an increased likelihood of concurrent *S. aureus* conjunctival and nasal colonisation, but it is unclear whether the bacteria in these reservoir sites are truly the causative pathogens of their bacterial keratitis [38]. Nouwen et al. demonstrated that host characteristics significantly influence the bacterial population of the nasal reservoir. Their findings determined that individuals who were non-carriers of *S. aureus* were likely to replace strains that had been inoculated, whereas persistent carriers were more likely to repopulate with their own strain after inoculation with different strains following antibiotic nasal ointment [45].

Furthermore, finding the causative pathogen often presents a problem with traditional ocular surface microbiological culture methods. It is estimated that the causative pathogen is correctly identified in only 4 in 10 cases of ocular infection with classical microbiological methods [46]. New approaches to the method of collecting corneal samples include the polytetrafluoroethylene (PTFE) impression membrane, which was found to have significantly better isolation rates compared to a surgical blade and was also less invasive [47]. The evolution of metagenomic next-generation sequencing may provide an effective diagnostic aid to finding the pathogen responsible for causing keratitis, and may better help us understand the differences in the ocular surface microbiome between healthy individuals and patients with keratitis or predisposing risk factors, such as ocular surface disease [46]. Metagenomic shotgun sequencing has also revolutionised the taxonomic profiling of microorganisms present in the corneal graft preservation medium, which is another source of infection that can predispose one to keratitis post-procedure, especially since topical corticosteroids are used to prevent graft rejection [48]. There are, however, some limitations of metagenomic next generation sequencing; it is expensive and time consuming, and relatively new in the field of ocular infections. Samples are prone to environmental contamination; for example, corneal samples can be contaminated during collection with bacteria from the eyelid. Specific limitations to the shotgun sequencing approach include the dominance of the human host background bacterial species within the sample, but this can be remedied to a certain extent by host depletion (decreasing host background) or targeted sequencing methods [49]. 

## 5. Current Recommendations in Management and Antibiotic Resistance

The Royal College of Ophthalmologists recommends that treatment of keratitis should begin with a broad spectrum antibiotic as empirical therapy [32], even before the culture and sensitivity results have returned [41]. Antibiotic therapy is usually topical monotherapy [50]. Fluoroquinolones are a commonly used group of antibiotics for empirical therapy in ophthalmology [51]; these include ciprofloxacin, moxifloxacin, levofloxacin and ofloxacin [32]. Corticosteroid drops are occasionally used as an adjunct to reduce epithelial inflammation [29]; however, there is little evidence of proven benefit [41]. The Steroids for Corneal Ulcers Trial by Srinivasan et al. showed no significant difference in best corrected visual acuity between patients receiving adjunct corticosteroid drops versus a placebo [52]. Response to treatment is evaluated within the first couple of days, and the treatment course is modified according to the culture results. For cases of multi-drug resistant Gram-positive bacterial keratitis such as MRSA, vancomycin is regarded as the best option [50]. Since the first case in 2002, there have been only 52 cases of vancomycin-resistant *S. aureus* reported worldwide [53]. Linezolid is another potential option for targeting multi-drug resistant Gram-positive bacteria in individuals with vancomycin intolerance [50]. In cases of unresponsive treatment, corneal rupture or recurrent keratitis, a corneal graft may be required.

Increased resistance towards fourth-generation fluoroquinolones has been observed, especially with *S. aureus* [20,37,51]. Vancomycin also has disadvantages such as toxicity and high cost [4]. There is a great abundance of research on antibiotic susceptibility of bacteria, especially with the rise of MRSA. Some studies conclude that MRSA has a higher overall resistance to multiple antibiotics compared to MSSA [4,54]. Others have found that MRSA is still susceptible to a variety of antibiotics such as tetracycline and gentamicin [10,55,56] and that progression to vancomycin is not always necessary [37]. Moreover, there is evidence to suggest that MSSA keratitis leads to similar visual outcomes as MRSA [37]. Nonetheless, most studies agree that the increasing resistance to fluoroquinolones is a significant concern [51], especially since they are the first line of treatment for unidentified bacterial keratitis. Research on novel antimicrobial therapies is extensive and covers many different aspects, such as contact lenses and cross-linking.

## 6. Novel Therapeutics

Variations in antibiotic therapy have been tested. A common approach is the use of dual therapy rather than monotherapy. Effective combinations for *S. aureus* include meropenem/ciprofloxacin [57], cefazolin/tobramycin [56], and polymyxin B-trimethoprim/rifampin [58]. There are other antibiotics not normally considered for ophthalmology that have been researched in the context of bacterial keratitis; examples include balofloxacin [59] and tigecycline [60,61]. Both were shown to be potential alternatives to vancomycin. 

Drug delivery to the anterior eye is challenging, especially since the cornea is avascular [62]. Other factors such as tear dilution can also prevent optimal concentrations from reaching the targeted area [63], and systemic side effects can result due to drainage of the drug into the lacrimal system [64]. Initial management of bacterial keratitis usually requires hourly drop instillation of antibiotics, which can be challenging to follow and a burden to patients [50]. An area of research for novel therapies concern methods that improve drug delivery to the ocular surface and consequently improve the performance of the antibiotic. Examples that are simple but effective are polysaccharides. Xanthan gum is a water-soluble polysaccharide, and one study has shown that a xanthan gum vehicle improves the bactericidal effects of a combination of tobramycin and dexamethasone on *S. aureus* [65]. Wu et al. also found beneficial effects of a polysaccharide isolated from the medicinal plant *Bletilla striata* on *S. aureus* using levofloxacin [66]. Polysaccharides specifically increase drug bioavailability by improving the drug contact time [67]. 

Microemulsions are another novel method of ocular drug delivery [64]. The drug is converted into tiny droplets 10 to 100 nm in diameter, covered in surfactant. Microemulsion delivery is effective because the structure of the cornea is represented by a lipid-water-lipid sandwich [68]. The outer layer of the cornea is a barrier to hydrophilic substances but is lipid-soluble; thus, microemulsions are able to effectively deliver a drug to the stroma. A combined in vivo and in vitro study has shown microemulsion delivery to be promising in rabbit keratitis [64]. Antibiotics can also be delivered to the eye in a similar manner by liposomes, a capsule made of a phospholipid bilayer. Liposomes also have the added benefit of being able to carry both hydrophilic and hydrophobic drugs [63]. Many studies have already demonstrated the potential of liposomes in increasing the corneal penetration and stability of ocular drugs [63]. Furthermore, Mishra et al. found that contact lenses equipped with liposomes are capable of providing a stable release of antibiotic over six days, which was effective against *S. aureus* in vitro [69]. Figure 4 illustrates the structural differences between microemulsions and liposomes.

Since improper contact lens hygiene remains a significant risk factor for bacterial keratitis [32], studies have focused on current disinfecting contact lens solutions [70] and novel options. One such example is the metal organic framework AGMNA, which includes silver (a known antimicrobial agent) and an anti-metabolite, 2-thio-nicotinic acid (H_2_MNA) [31]. This compound has been investigated as a substitute for the more widely used component polyhexanide, for which there are concerns due to its suspected carcinogenic effects [71]. AGMNA demonstrated higher effectiveness than polyhexanide against *S. aureus*, despite showing no signs of in vivo or in vitro toxicity [72]. 

In addition to disinfecting solutions, antimicrobial drugs and compounds have been incorporated into the design of the contact lens themselves. AGMNA has proven to be a good candidate for the design of a contact lens with an inherent antimicrobial effect against *S. aureus*, *P. aeruginosa* and *S. epidermidis* [31]. Contact lenses have also been modified to slowly release antimicrobials onto the cornea, such as nitric oxide [73] and ciprofloxacin [74] over 8–16 h. These types of lenses have been proven to be effective against *S. aureus* and *P. aeruginosa* in ex vivo rabbit corneal models [75]. Moreover, the slow release provided by the lenses would reduce side effects that are often seen with topical treatments.

Nanoparticles have also been investigated outside of the contact lens area of research. Gelatin-capped silver nanoparticles have been shown to not only have antimicrobial activity, but also an antiangiogenic effect in vitro [76]. This added benefit is significant in keratitis where inflammation can result in the release of angiogenic factors and subsequent corneal neovascularisation. Nanoparticles can also be used to load antibiotics; moxifloxacin nanoparticles show increased corneal penetration. An in situ gel is a liquid that turns into a gel upon contact with the cornea and increases the bioavailability of a loaded drug. Upadhyay et al. have shown that moxifloxacin nanoparticles can be effectively combined with an in situ gel as a promising novel therapy [77]. Another breakthrough in nanoparticle research is molecular imprinting. It is possible to use template molecules to convert nanoparticles into the synthetic equivalent of antibodies; this was used to successfully target lipopolysaccharides, a known disease severity marker for *P. aeruginosa*, in a keratitis model [78]. This approach could be used similarly to target MRSA with key disease markers.

Antimicrobial peptides form part of the innate immune system. Peptides remain a viable option, especially with uprising antibiotic resistance; however, there are still concerns with toxicity and stability [79]. There are a wide variety of different peptides that have been investigated for numerous different keratitis-causing organisms. Table 1 summarises novel peptides discovered within the last five years that have shown to be effective against *S. aureus*, including MRSA, during ocular infection.

The novel treatments discussed thus far have been based on medical therapy. A potential alternative to medical therapy is corneal cross-linking (CXL). CXL is a typical treatment for keratoconus and corneal ectasias [84] that consists of a combination of an ultraviolet-A (UVA) beam and a photosensitising agent riboflavin. It is typically used to generate free radicals, forming chemical bonds between collagen fibrils of the stroma. Although keratitis is a severe complication of cross-linking [40], it was also discovered that the UVA and riboflavin involved in the cross-linking procedure may have potential antimicrobial effects; the free radicals are also capable of damaging bacterial DNA [85]. CXL treatment for keratitis is sometimes referred to as photoactivated chromophore for keratitis-corneal cross-linking (PACK-CXL) [86]. In vitro studies have demonstrated the effectiveness of CXL against pathogens such as *S. aureus* [86], even with other photosensitising agents such as rose bengal [87] and toluidine blue [88]. One recent study concluded that CXL used in combination with topical antibiotics such as tobramycin can produce synergistic effects [89]. A topical CXL approach has also been developed, using sodium hydroxymethylglycinate as a cross-linking agent instead of using the UVA/riboflavin photochemical technique. This topical approach proved effective against MRSA and MSSA in vitro; its advantages include avoiding the need to remove the epithelium or expose the lens or retina to harmful UV light [90]. One concern over the use of CXL is a severe post-operative complication known as corneal melt, which can also occur due to keratitis itself. Whilst some initial clinical studies showed a potential benefit of the application of CXL in preventing corneal melt [91,92,93], more recent studies have shown that the outcomes of CXL for keratitis are uncertain, with reports of corneal melt occurring after CXL [94,95]. Thus, further clinical validation is required for CXL and keratitis.

Plasma and phage therapy are examples of even more novel and innovative treatment options for bacterial keratitis. Plasma is an ionised gas capable of exhibiting antimicrobial properties via its ability to produce reactive oxygen species; it also exhibits wound healing and anti-inflammatory properties [96]. With plasma therapy, safety within the cornea is the main focus [2]. Reitberger et al. has demonstrated that argon-based plasma therapy could be successfully used in conjunction with antibiotics [96], but there is a need for further clinical validation. Phage therapy (Figure 5) involves using a viral bacteriophage to infect and kill bacteria. There have only been a few clinical studies specifically involving keratitis; only one study has shown phage therapy to be effective against *P. aeruginosa* keratitis in mice [97]. There has also been a reported case of a patient recovering from MRSA keratitis after phage therapy [98]. The effectiveness of phage therapy against a wide number of different non-ocular bacterial colonies has been confirmed by other studies [99], but there is a need for further investigation focusing specifically on *S. aureus* keratitis isolates. 

## 7. Conclusions

The problem of antibiotic resistance has escalated significantly. Epidemiological studies have shown *S. aureus* to be one of the organisms most commonly cultured from bacterial keratitis. In addition, *S. aureus* already possesses a fearsome reputation for its resistance to antibiotics. This review has highlighted a wide variety of methods to tackle the problem by discussing new antimicrobial substances and other novel treatment modalities, such as drug-releasing contact lenses and PACK-CXL. All have shown effectiveness against bacterial keratitis, but most are restricted to animal or in vitro models. Ultimately, further clinical validation with human participants is required as the next step.

## Figures and Tables

**Figure 1 jcm-10-00758-f001:**
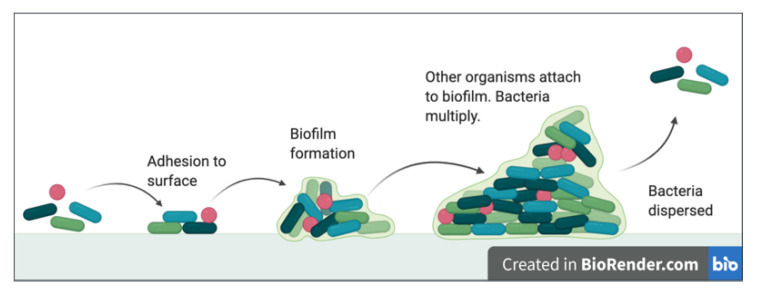
A diagram illustrating biofilm formation. *Adapted from “Polymicrobial Biofilm”, by BioRender.com (2021). Retrieved from https://app.biorender.com/biorender-templates (accessed on 9 February 2021).*

**Figure 2 jcm-10-00758-f002:**
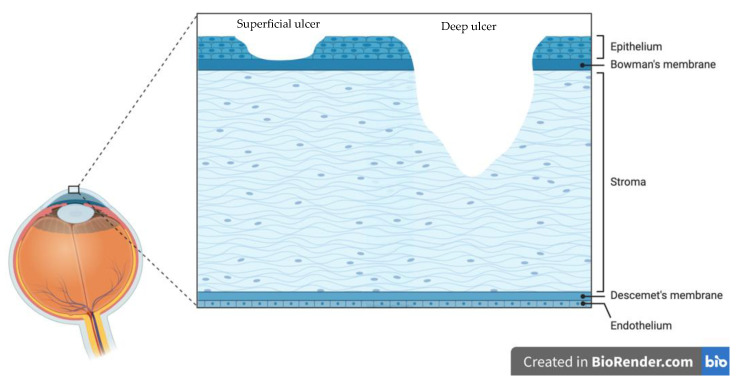
A simplified diagram showing the various layers of the cornea affected by a superficial and a deep corneal ulcer. *Adapted from “Corneal Anatomy”, by BioRender.com (2021). Retrieved from https://app.biorender.com/biorender-templates (accessed on 9 February 2021).*

**Figure 3 jcm-10-00758-f003:**
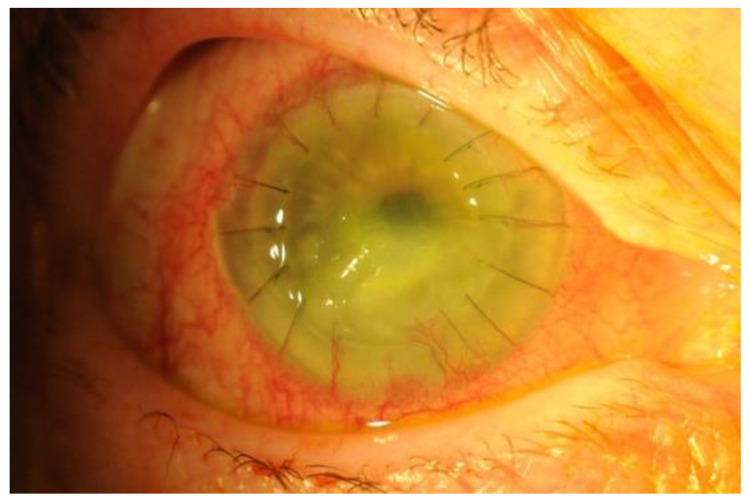
*S. aureus* keratitis in a patient with previous penetrating corneal transplant.

**Figure 4 jcm-10-00758-f004:**
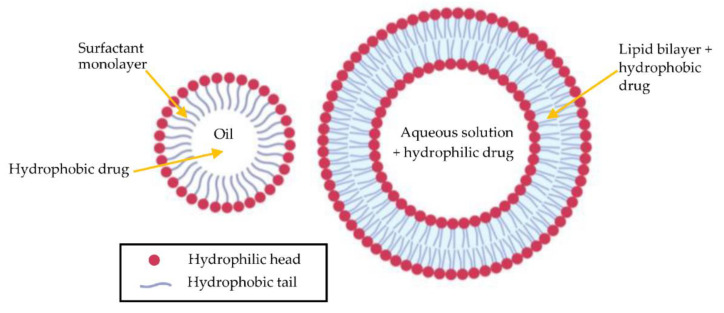
Structural differences between an oil-in-water microemulsion (**left**) and liposome (**right**). *Created with BioRender.com.*

**Figure 5 jcm-10-00758-f005:**
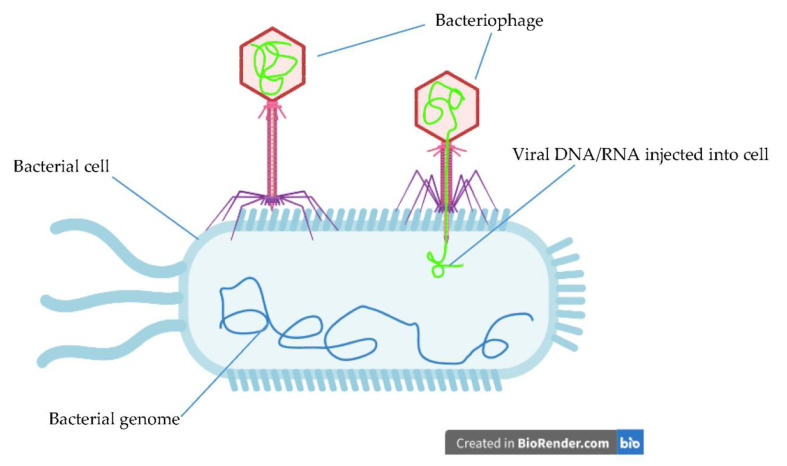
Infection of a bacterial cell with a bacteriophage. *Created with BioRender.com.*

**Table 1 jcm-10-00758-t001:** Novel antimicrobial peptides that are effective against *S. aureus* keratitis.

Peptide/Protein Name	Classification	Mechanism of Action	Study
LyeTxI-b	Synthetic peptide derived from *Lycosa erythrognatha* spider venom	Alters permeabilisation and forms pores within bacterial membrane	Silva et al. (2019) [7]
D-Arg_4_-Leu_10_-Teixobactin	Analogue of teixobactin, a cyclic depsipeptide	Binds to pyrophosphate motifs of bacterial cell-wall substrates, such as lipid II (precursor of peptidoglycan), and lipid III (precursor of teichoic acid)	Parmar et al. (2018) [80]
RP442, RP443, RP444	Designed host-defence peptide	Electrostatic interactions with bacterial membrane	Clemens et al. (2017) [81]
poly-ε-lysine	Biosynthetic polymer	Induces a loss in membrane potential	Venkatesh et al. (2017) [82]
Brilacidin (PMX30063)	Defensin mimetic	Membrane depolarisation	Kowalski et al. (2016) [83]

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
