# Peer review of "Staphylococcus aureus Keratitis: Incidence, Pathophysiology, Risk Factors and Novel Strategies for Treatment"

_jcm, 2021, doi:10.3390/jcm10040758_

Round 1

Reviewer 1 Report

This manuscript titled, “Staphylococcus aureus keratitis: pathophysiology and novel strategies for treatment” chronicles this bacterial keratitis condition in the eye and effectively summarizes the disease physiology, and presents current and future therapies for the disease. The reviewer finds the manuscript quite informative and believes that it will be an important resource to the bacterial keratitis research community.

Suggestion:

  • As this manuscript is broken down into incidence, pathophysiology, risk factors and treatments, the reviewer suggests that the title be edited to reflect the sections presented in the manuscript.  

Author Response

This manuscript titled, “Staphylococcus aureus keratitis: pathophysiology and novel strategies for treatment” chronicles this bacterial keratitis condition in the eye and effectively summarizes the disease physiology, and presents current and future therapies for the disease. The reviewer finds the manuscript quite informative and believes that it will be an important resource to the bacterial keratitis research community. 

Suggestion:

  • As this manuscript is broken down into incidence, pathophysiology, risk factors and treatments, the reviewer suggests that the title be edited to reflect the sections presented in the manuscript.  

Authors >> We’ve changed the title from “pathophysiology and novel strategies for treatment” to “incidence, pathophysiology, risk factors and novel strategies for treatment”

Reviewer 2 Report

This review thourougly assesses Staphyloccus Aureus keratitis. The manuscript is well written and presents novel methods for the management of bacterial keratitis and potential future developments in the field. The manuscript is suitable for publication after a minor revision regarding two points.

  1. Page 4, line 160.  Next generation sequencing is a relatively new development in the field of ocular infections. The authors should also highlight the limits of this technology. 
  2. Page 7 line 267. Different studies have reported results of Corneal collagen cross-linking for management of keratitis with heterogeneous results. Severe complications such as post-op corneal melting have been reported and these should be cited.  Recent studies and reviews highlight that results for CXL in bacterial keratitis are uncertain (Cochrane review Jun 2020 + Clair protocol recently published in the Cornea Journal

Author Response

Reviewer 2

This review thourougly assesses Staphyloccus Aureus keratitis. The manuscript is well written and presents novel methods for the management of bacterial keratitis and potential future developments in the field. The manuscript is suitable for publication after a minor revision regarding two points.

  1. Page 4, line 160.  Next generation sequencing is a relatively new development in the field of ocular infections. The authors should also highlight the limits of this technology. 

Authors >> We have included this point and added the limitations of both metagenomic NGS and specifically shotgun sequencing.

  1. Page 7 line 276. Different studies have reported results of Corneal collagen cross-linking for management of keratitis with heterogeneous results. Severe complications such as post-op corneal melting have been reported and these should be cited.  Recent studies and reviews highlight that results for CXL in bacterial keratitis are uncertain (Cochrane review Jun 2020 + Clair protocol recently published in the Cornea Journal

Authors >> Thank you for this feedback. We have highlighted the concerns of corneal melt as a complication reported in more recent studies as a contrast to earlier studies showing a potential benefit in prevention. We have discussed the suggested manuscripts.

Reviewer 3 Report

Lee, Somerville et al. provide an interesting update on S. aureus keratitis to include its pathophysiology, epidemiology and treatments. I would consider this a valid and worthwhile contribution to vision science, and it is accessible to most readers with any medical training.

I do have several comments. Many are minor (or "style") suggestions, others are more editorial and will require either journal oversight or another review by me to check for appropriate changes.

OVERALL: This manuscript is a well-written contribution. It is a little "folksy" in places, which I will point out in the "MINOR" section below. It sufficiently reviews S. aureus, particularly exploring medical treatments and one surgical intervention.

MINOR:

  1. Line 13: "...a lot of work..." seems folksy to me. Suggest, "Much has been previously reported..."
  2. Line 84/157/etc.: Semi-colons are overused in this manuscript. I believe the authors are adhering to standard English in doing so, but I have a North American bias. They are exhausting after two or three of them. (The one in Line 220 may not even be correct.) Please consider reformatting some of the phrases using ";"

MAJOR:

  1. Line 9: Please spell out S. and P., as it is the first reference to them.
  2. Lines 123-127: These sentences do not fit very well and do not read well. I understand that cross-linking is a risk, but does DM further mediate the risk after cross-linking or is it an independent risk? Atopic conjunctivitis and other surface disease has been shown to be an independent risk. Please clarify the intent of lines 126-7.
  3. Similarly, lines 128-132 do not read as well as they could. Consider moving the last sentence earlier in the paragraph.
  4. Line 159: The authors should probably spell out polytetrafluoroethylene the first time.
  5. Line 182: The reference (Gupta et al.) is from 2011 and probably needs updating. I wager the number is higher now.
  6. Lines 228-229: Why all this space?
  7. Lines 231 and 234: Need references for these statements.
  8. Line 232: It would be too much to write out Ag6[etc...] for AGMNA. However, readers would benefit from some additional context here. I suggest the authors see Rossos et al., Mat Sci & Engr, Vol 111, 2020 for that context. (There are many others.)
  9. Line 268: "CXL" generally mean collagen cross-linking. In ophthalmology, it means corneal cross-linking. I would spell out which one is meant the first time.

Author Response

Lee, Somerville et al. provide an interesting update on S. aureus keratitis to include its pathophysiology, epidemiology and treatments. I would consider this a valid and worthwhile contribution to vision science, and it is accessible to most readers with any medical training.

I do have several comments. Many are minor (or "style") suggestions, others are more editorial and will require either journal oversight or another review by me to check for appropriate changes.

OVERALL: This manuscript is a well-written contribution. It is a little "folksy" in places, which I will point out in the "MINOR" section below. It sufficiently reviews S. aureus, particularly exploring medical treatments and one surgical intervention.

MINOR:

  1. Line 13: "...a lot of work..." seems folksy to me. Suggest, "Much has been previously reported..."

Authors >> We have followed your suggestion and changed this, thank you.

  1. Line 84/157/etc.: Semi-colons are overused in this manuscript. I believe the authors are adhering to standard English in doing so, but I have a North American bias. They are exhausting after two or three of them. (The one in Line 220 may not even be correct.) Please consider reformatting some of the phrases using ";"

Authors >> We have changed the semicolon on lines 220, 86, 155 and 160.

MAJOR:

  1. Line 9: Please spell out S. and P., as it is the first reference to them.

Authors >> We have corrected this.

  1. Lines 123-127: These sentences do not fit very well and do not read well. I understand that cross-linking is a risk, but does DM further mediate the risk after cross-linking or is it an independent risk? Atopic conjunctivitis and other surface disease has been shown to be an independent risk. Please clarify the intent of lines 126-7.

Authors >> Thank you for pointing this out, we have clarified and restructured those lines. The intent was to show that atopy and DM increase risk of keratitis after corneal cross linking. I have mentioned earlier that ocular surface disease is also an independent risk.

  1. Similarly, lines 128-132 do not read as well as they could. Consider moving the last sentence earlier in the paragraph.

Authors >> We have restructured the paragraph so that these sentences are earlier and taken a new paragraph for risk factors of keratitis post-CXL.

  1. Line 159: The authors should probably spell out polytetrafluoroethylene the first time.

Authors >> We have included this

  1. Line 182: The reference (Gupta et al.) is from 2011 and probably needs updating. I wager the number is higher now.

Authors >> Cong et al. (2020) reports that there have been 52 cases since 2002. We have included this reference instead of Gupta et al.

  1. Lines 228-229: Why all this space?

Authors >> We have removed this space.

  1. Lines 231 and 234: Need references for these statements.

Authors >> We have included more references for statements in that paragraph.

  1. Line 232: It would be too much to write out Ag6[etc...] for AGMNA. However, readers would benefit from some additional context here. I suggest the authors see Rossos et al., Mat Sci & Engr, Vol 111, 2020 for that context. (There are many others.)

Authors >> We have provided some further clarification on what AGNMA is and included Rossos et al. as a reference.

  1. Line 268: "CXL" generally mean collagen cross-linking. In ophthalmology, it means corneal cross-linking. I would spell out which one is meant the first time.

Authors >> Thank you for this. We have mention cross linking in two main areas in the manuscript that are quite spaced out therefore at the beginning of each area I have spelt out the full name corneal cross-linking followed by abbreviation (CXL). This is to help the reader in-case they forget what CXL means.

Reviewer 4 Report

Overall, this is a well-written review on S. aureus keratitis that takes the reader on a logical progression from epidemiology to pathophysiology to clinical presentation, etc. Also, each section is nicely integrated into the next. This reviewer is a corneal specialist, not a microbiologist or immunologist, and I found it very informative with just enough of a review to provide the essential information and references to follow for additional detail. Few specific comments:

  1. A picture of S. aureus keratitis would be helpful for the general audience and perhaps a diagram of the cornea in cross-section depicting the layers of the cornea involved in corneal ulcers.
  2. In the section on risk factors for S. aureus keratitis, the role of ocular surface disease is mentioned. Ocular surface disease is a broad range of conditions. Is there any specific increased risk in the presence of eyelid margin disease (meibomian gland dysfunction, blepharitis)?
  3. Line 176: the SCUT trial would be a good reference here to incorporate the role of corticosteroids in corneal ulcers: Srinivasan M et al. Arch Ophthalmol. 2012 Feb;130(2):143-50.
  4. Line 180: If linezolid should be used, why is vancomycin the gold standard?
  5. Line 181: There are more than 20 cases of vanco resistant S. aureus worldwide. Perhaps the 20 reflects the number in the literature.
  6. Line 203: Might be worth mentioning here the burden of hourly drop instillation for initial management of corneal ulcers.
  7. Line 214: When the corneal epithelium is disrupted (or absent) in keratitis, the lipid-water-lipid sandwich is disrupted. Does that affect the activity of microemulsions?

Author Response

Overall, this is a well-written review on S. aureus keratitis that takes the reader on a logical progression from epidemiology to pathophysiology to clinical presentation, etc. Also, each section is nicely integrated into the next. This reviewer is a corneal specialist, not a microbiologist or immunologist, and I found it very informative with just enough of a review to provide the essential information and references to follow for additional detail. Few specific comments:

  1. A picture of S. aureus keratitis would be helpful for the general audience and perhaps a diagram of the cornea in cross-section depicting the layers of the cornea involved in corneal ulcers.

Authors >> Thank you for the suggestion. We have included the diagram of the corneal layers as figure 2 and add an original image of staph aureus keratitis myself.

  1. In the section on risk factors for S. aureus keratitis, the role of ocular surface disease is mentioned. Ocular surface disease is a broad range of conditions. Is there any specific increased risk in the presence of eyelid margin disease (meibomian gland dysfunction, blepharitis)?

Authors >> We have clarified this by adding a statement of what ocular surface diseases specifically increase risk. We discussed about ocular surface disease in general, even if there is not strong evidence that eyelid disease conferring greater risk than other types of ocular surface disease for S. aureus keratitis.

  1. Line 176: the SCUT trial would be a good reference here to incorporate the role of corticosteroids in corneal ulcers: Srinivasan M et al. Arch Ophthalmol. 2012 Feb;130(2):143-50.

Authors >> Thank you, we have included the SCUT trial as a reference.

  1. Line 180: If linezolid should be used, why is vancomycin the gold standard?

Authors >> We corrected our statements, we have amended them.

  1. Line 181: There are more than 20 cases of vanco resistant S. aureus worldwide. Perhaps the 20 reflects the number in the literature.

Authors >> This point was brought up also by reviewer 2. We corrected, and the total now is 52 according to Cong et al

  1. Line 203: Might be worth mentioning here the burden of hourly drop instillation for initial management of corneal ulcers.

Authors >> we have mentioned this, thank you.

  1. Line 214: When the corneal epithelium is disrupted (or absent) in keratitis, the lipid-water-lipid sandwich is disrupted. Does that affect the activity of microemulsions?

Authors >> The epithelium is the most important barrier to overcome for drug delivery, and the stroma is still able to allow lipid-soluble drugs through, just less effectively than water-soluble drugs. When the epithelium is disrupted then the stromal drugs uptake is easier.